# Application of an Artificial Neural Network to Identify the Factors Influencing Neurorehabilitation Outcomes of Patients with Ischemic Stroke Treated with Thrombolysis

**DOI:** 10.3390/biom13020334

**Published:** 2023-02-09

**Authors:** Marco Iosa, Stefano Paolucci, Gabriella Antonucci, Irene Ciancarelli, Giovanni Morone

**Affiliations:** 1Department of Psychology, Sapienza University of Rome, 00185 Rome, Italy; 2SmArt Lab, IRCCS Santa Lucia Foundation, 00179 Rome, Italy; 3Department of Life, Health and Environmental Sciences, University of L’Aquila, 67100 L’Aquila, Italy

**Keywords:** cerebrovascular accident, brain, injury, rehabilitation, machine learning, artificial intelligence

## Abstract

The administration of thrombolysis usually reduces the risk of death and the consequences of stroke in the acute phase. However, having received thrombolysis administration is not a prognostic factor for neurorehabilitation outcome in the subacute phase of stroke. It is conceivably due to the complex intertwining of many clinical factors. An artificial neural network (ANN) analysis could be helpful in identifying the prognostic factors of neurorehabilitation outcomes and assigning a weight to each of the factors considered. This study hypothesizes that the prognostic factors could be different between patients who received and those who did not receive thrombolytic treatment, even if thrombolysis is not a prognostic factor per se. In a sample of 862 patients with ischemic stroke, the tested ANN identified some common factors (such as disability at admission, age, unilateral spatial neglect), some factors with higher weight in patients who received thrombolysis (hypertension, epilepsy, aphasia, obesity), and some other factors with higher weight in the other patients (dysphagia, malnutrition, total arterial circulatory infarction). Despite the fact that thrombolysis is not an independent prognostic factor for neurorehabilitation, it seems to modify the relative importance of other clinical factors in predicting which patients will better respond to neurorehabilitation.

## 1. Introduction

Thrombolysis is an intravenous medical procedure for treating specific subjects with ischemic stroke [1]. It is a drug based on recombinant tissue plasminogen activator (rt-PA, usually alteplase or tenecteplase) that is injected in the early acute phase to disperse the clot, restore blood supply to the brain, and reduce the risk of death [1].

Guidelines and authorizations for the thrombolytic treatment administration are quite different among countries: in Europe its, use is recommended within 4.5 h of the acute event, with cautions for severe stroke and people older than 80 years (in the USA, the limit of time was 3 h, without the above recommended cautions) [1].

Previous studies about thrombolysis were focused on the safety and efficacy of this intervention in the acute phase, analyzing the probability of surviving and reducing disability. Surprisingly, there is a lack of knowledge about the long-term recovery patterns and service requirements of people treated with thrombolysis, both of which have significant implications for patient management by rehabilitation teams [2].

To understand this marginal prognostic role of thrombolysis in neurorehabilitation, it should be considered that patients needing intensive therapies after having received thrombolytic intervention are those for whom thrombolysis was only partially effective in reducing the sequelae of stroke. In fact, among the wide body of literature investigating the prognostic factors related to the positive outcome of neurorehabilitation in patients with stroke, thrombolysis is often absent or not statistically significant. These studies identified the following as the most common factors: age, severity of stroke in terms of disability, presence of unilateral spatial neglect, aphasia, and total anterior circulation infarct [3,4,5].

These prognostic studies are helpful for promptly informing the patients, their caregivers, and family members, for planning the post-hospital discharge phase, and for adequately allocating the proper resources [6,7]. The recent development of artificial intelligence (AI) could provide a powerful tool for a deeper analysis of prognostic factors using a machine learning approach [8,9,10,11]. The main difference with classical statistical approaches is that artificial neural networks assign a weight to each of the investigated factors. Hence, AI may provide a more complex, but also a more detailed, picture to understand the variables influencing the neurorehabilitation outcome in a heterogeneous population such as that of patients with stroke [5].

Furthermore, the classical statistical approach based on regressions might have hidden the role played by thrombolysis in neurorehabilitation because its effects could be combined with and/or masked by the severity of stroke. In fact, thrombolysis may reduce the level of remaining disability at admission into a neurorehabilitation hospital, a factor usually reported as the most important prognostic one.

In this study, we used an artificial neural network (ANN) to analyze the prognostic factors of neurorehabilitation outcome in the subacute phase, separating patients who received thrombolytic intervention in the acute phase from those who did not.

The aim was to test, using a machine-learning approach, the hypothesis that thrombolysis could change the weights of prognostic factors for neurorehabilitation in patients with stroke.

## 2. Materials and Methods

### 2.1. Patients

This study was a secondary analysis conducted on a large database already used in some previous studies [3,5,12,13] and further augmented with new data. The inclusion criteria were: diagnosis of ischemic stroke confirmed by brain imaging (magnetic resonance imaging or computerized tomography), subacute phase of stroke, and admission to a neurorehabilitation hospital. The exclusion criteria were: previous cerebrovascular accidents, hemorrhagic stroke, subarachnoid hemorrhage, and presence of other chronic disabling pathologies (i.e., severe Parkinson’s disease, polyneuropathy, cancer, or limb amputation).

Because our hospital is also an institute of research, at the time of admission all patients signed an informed consent for the utilization of their data in translational research. In the present study, a sample of 862 patients was extracted from the dataset according to the above inclusion/exclusion criteria and further divided into a subgroup previously treated with thrombolysis (TG) and another not treated with it (NTG).

The dataset reported for each patient consisted of 22 variables accounted for at admission to the neurorehabilitation hospital. The variables were as follows: age (continuous variable), time (days) between the stroke acute event and admission into the neurorehabilitation hospital (DAS, continuous variable), Barthel Index score (BI) at admission (ordinal variable, Barthel Index is a clinical scale assessing the independency of a patient into the activities of daily living), and binary variables such as gender, if patients received thrombolysis, damaged hemisphere, if there was a diagnosis of hypertension, heart diseases, diabetes, depression, epilepsy, dysphagia, malnutrition, obesity, Broca’s aphasia (related to deficits in speech and language production), Wernicke’s aphasia (related to deficits in language understanding), global aphasia (including both the previous types of language deficits), unilateral spatial neglect (USN, related to deficits in reporting or responding to stimuli presented from the space contralateral to the lesion, often a right hemisphere lesion), and the category of Bamford classification. This latter variable refers to the anatomical type of stroke and was further divided into four binary variables, in accordance with previous studies that dichotomized each one of these categories [3,5]: TACI (total anterior circulatory infarction), PACI (partial anterior), POCI (partial posterior), and LACI (lateral anterior).

The dependent variable was the outcome: good responders were defined as subjects who were discharged from a neurorehabilitation hospital with a BI-score >75, whereas low-medium responders were those who died, were transferred to emergency hospitals, or were discharged with a BI-score ≤75.

### 2.2. Neurorehabilitation

Our neurorehabilitation ward is part of a hospital for subacute rehabilitation, and it is formed by a wide gym, specific rooms for individual treatments, and bedrooms with two beds each. The neurorehabilitation was planned for each patient by a pool of neurologists and physiatrists and administered by therapists 6 days a week, in 3 sessions per day, with each session lasting 1 h. According to the needs of the patient, individual therapy could include physical therapy, cognitive therapy, neglect or speech therapy, occupational therapy, and specific therapy for swallowing, bowel, and bladder dysfunctions. All rehabilitation treatments began within 24 h of admission. Physiotherapy and language treatment continued throughout the hospital stay, and the training for neglect lasted 8 consecutive weeks.

### 2.3. Artificial Neural Network

The Artificial Neural Network analysis was conducted by the ARIANNA model (ARtificial Intelligent Assistant for Neural Network Analysis), already used in previous studies [5,14,15,16]. This model is a multilayer perceptron procedure composed of an input layer (from which the variables listed above entered), two hidden layers (each with five), and a final output layer (the predictor of the dependent variable). The architecture of the ANN was that of a Feed Forward Neural Network (FFNN), with data moving only in one direction, from the input nodes through the two hidden layers to the output node. The activation function for all the units in the hidden layers and for the output layer was a hyperbolic tangent. The chosen computational procedure was based on online training [14,15]. The ANN worked on half of the data (training phase), then tested the other half for (testing phase). The procedure uses random number generation during the random assignment of partitions for subsampling cases between training and testing. The ANN was developed in the Statistical Package for the Social Sciences software (SPSS, version 23.0, IBM, Chicago, IL, United States).

### 2.4. Statistical Analysis

Continuous and ordinal variables have been reported in terms of mean and standard deviation, compared using the Mann–Whitney U-test, and then dichotomized for prognostic analyses as follows: age < or ≥65 years, BI-score at admission ≤ or >20, DAS ≤ or >14 days. Binary variables have been compared using the chi-squared test.

The ANN results are reported in terms of accuracy (percentage of correct classification of all cases), sensitivity (percentage of correct classification of good responders), specificity (percentage of correct classification of medium-low responders), area under the receiver operating characteristic (ROC) curve, percentage weight of each factor, and relevant standard errors of three ANN applications.

A binary forward stepwise logistic regression analysis was also conducted to identify, among the analyzed factors, those significantly associated with a positive outcome to allow a comparison with ARIANNA results. Values of coefficient Beta, as well as their exponential values corresponding to the odds ratio (OR), were computed and associated with their relevant *p*-values (statistically significant if <0.05).

All the statistical analyses, as well as the ANN, have been performed using the Statistical Package for the Social Sciences software (SPSS, version 23.0, IBM, Chicago, IL, United States).

## 3. Results

Of the 862 analyzed patients, 140 (16%) received a thrombolytic intervention. The demographical and clinical data are shown in Table 1 for the whole sample as well as for the TG and NTG subgroups. The statistically significant differences between these two subgroups are limited to age (TG was about 3.6 years younger than NTG), Bamford Classification (with higher percentages of TACI and PACI in TG than in NTG), and cases of death that did not occur in TG, whereas 2.6% of NTG patients died. 

The accuracy obtained by ANN was quite similar among the analyses performed on the whole sample and those performed on the two subgroups (Table 2). The area under the curve was also similar between TG and NTG. In general, the sensitivity of identifying good responders was much lower than the specificity of identifying medium–lower responders. 

As shown in Figure 1, the most important factor for the whole sample, as well as for TG and NTG, was the Barthel Index score at admission. The weight associated with age ranged between 5.5% and 6.2% in the three groups. Thrombolysis accounted for a relative weight of only 1.8%. However, important differences could be noted between TG and NTG. The main difference was related to the BI-score at admission, which weighted 30.6% in NTG and about half (15.7%) in TG. This reduction in normalized importance increased the relative weight of other factors in TG. As shown in Figure 1, the factors with the largest difference between TG and NTG were: hypertension (+4.3% in TG), global (+3.2%), Wernicke’s aphasia (+3.4%), presence of epilepsy (+2.3%), obesity (+1.9%), and heart disease (+1.5%), together with a different distribution of Bamford classification (LACI: +6.8%).

Interestingly, all patients with global aphasia in TG (N = 23) had a poor outcome, whereas obesity was a prognostic factor for a positive outcome.

All these factors accounted for a difference of 23.4% with respect to NTG, whereas the difference in the BI-score at admission was 14.9%. This gap is explained by the lower weight in TG than in NTG and a number of other factors: the time between stroke and admission (−3.1%), the presence of dysphagia (−2.8%), and malnutrition (−0.9%).

Because BI-admission heavily weights the results of ANN, a recursive confirmatory analysis was performed after removing this factor from the input variables. Re-running the classification on the whole sample, the factors that accounted for a percentage of importance >5% were: global aphasia (16%), USN (13%), time from stroke (12%), age (11%), dysphagia (9%), TACI (9%), and Wernicke’s aphasia (5%). These results substantially confirmed those of the primary analysis.

### Comparison between ANN and Regression Analysis

To confirm or not the findings of the ANN, a classical binary logistic regression was performed for the whole sample, for TG, and for NTG. The results were partially similar to those obtained with ANN, with thrombolysis remaining out of the model for the whole sample (*p* = 0.664). Then, BI-score at admission, age, and TACI classification consistently resulted in the significant prognostic factors common to the three groups (Table 3).

Similarly, to the findings obtained by ANN, DAS and dysphagia entered the predictive model of NTG but not that of TG (as well as strokes located in the right vs. left hemisphere, not accounted for by ANN). Epilepsy and LACI entered the TG model and not the NTG model, in accordance with the findings of ANN. Hypertension and heart diseases remained out of the model, but with lower *p*-values for TG than for NTG, reflecting the results of ANN. In addition, malnutrition remained out of the model, but with a higher *p*-value for TG than NTG. The most significant difference was observed for global aphasia, which entered the NTG model and not the TG model, contrary to ANN’s suggestion. 

## 4. Discussion

The accuracy of our ANN was similar among subgroups and also with respect to that of regression analysis. The sensitivity in identifying the good responders was quite lower than the specificity in identifying the medium-low responders, probably because of the slightly lower number of patients classified as good responders (about 40% vs. 60%). Our ANN accuracy of about 85% falls into the range of 74–95% reported in a recent review about the use in stroke rehabilitation of different machine learning approaches (including random forest, gradient boosting, support vector machines, decision trees, and k-nearest neighbors) [11].

The regression analyses mainly confirmed the results of ANN. The former identified seven significant variables for the whole sample and NTG, similarly to the six variables with a weight >5% identified by ANN for these groups. However, ANN identified nine variables with a weight >5% for TG, providing a more complex picture than that described by the only five significant factors of regression analysis. 

Some variables had a quite consistent impact on the outcome between the subgroups, such as age, the weight of which ranged between 5.5 and 6.8% in ANN, confirmed by logistic regression and also in accordance with scientific literature [3,4,5]. Age was a common factor in both TG and NTG. However, as shown in Table 1, patients who received thrombolytic treatment were slightly younger, but statistically significantly younger, than patients of NTG. It is known that the probability of receiving intravenous thrombolysis for the treatment of stroke declines with increasing age [17]. Furthermore, strokes in younger age groups were commonly associated with cardiac diseases and partially with hypertension [18]. These comorbidities may be more important in patients receiving thrombolysis, not because of the treatment itself, but because thrombolysis is more commonly administered to patients with these cardio-circulatory deficits, and they may also play a role during neurorehabilitation. Vascular diseases have also been associated with epilepsy [19], and it could also explain some other differences highlighted by the comparison between TG and NTG. In general, brain circulation and brain pressure are two fundamental factors both for the risk of having a stroke and for the implications during the subacute and chronic phases, including the risk of a secondary stroke. Hence, the correct management of hypertension is mandatory for the primary prevention of stroke recurrences and also during the stroke neurorehabilitation program. Our findings indicated that this factor was particularly important in patients who received thrombolytic treatment. Clinicians may lower blood pressure, but at the same time it is necessary to ensure adequate perfusion of the brain during neurorehabilitation. The optimal balance between these two aspects should be modulated, also to favor neuroplasticity, on the basis of the type of stroke, comorbidities [20], and, as suggested by our results, also considering if the patient received thrombolytic treatment. It should be considered that our study excluded patients who had a hemorrhage after thrombolysis in the acute phase; further studies should investigate this aspect and potential hemorrhagic complications in people treated with thrombolysis. 

Unilateral spatial neglect resulted in a prognostic factor for ANN, whereas it remained out of the linear regression model, conceivably because it was masked by the role played by damage to the right hemisphere. According to ANN results, the scientific literature reports USN as a prognostic factor of neurorehabilitation outcome [4,21].

Shorter time from stroke and admission to a neurorehabilitation hospital, absence of dysphagia, and absence of malnutrition had greater weight as prognostic factors of a positive outcome in NTG than in TG. Conversely, in TG, other factors having greater weights with respect to NTG were LACI and obesity.

Furthermore, the effect of global aphasia was higher in TG for the ANN but associated with a higher OR for NTG by regression analysis. This difference could be due to the two different methodologies: the regression took into account only the minimum number of factors significant for the model, leaving out factors such as malnutrition and obesity, whereas the ANN considered all the factors, assigning them different weights [5]. The strict time-based approach of thrombolysis may restrict treatment options for patients with stroke symptoms involving aphasia because of their difficulties in describing their symptom onset [22]. In general, the role of global aphasia as a prognostic factor for a poor outcome in rehabilitation is studied and linked to the reduced capacity of the patient to understand and follow the instructions given during therapy sessions.

According to the literature, the effect of LACI on TG could be easily explained as a different classification from the most severe TACI, which played a common negative role for TG and NTG [5].

Malnutrition was found to be a negative prognostic factor for NTG and obesity was found to be a good prognostic factor for TG. These results are in line with the so-called obesity paradox [23], which was recently defined as the role of overweight and obesity in health preservation as a result of malnutrition and weight loss during hospitalization [24]. 

These results highlight the merit of the present study to investigate how thrombolysis may affect the other prognostic factor for functional recovery after neurorehabilitation. It is important to understand the prognosis of stroke patients undergoing thrombolysis in order to better target neurorehabilitation resources. Following the emerging approach of personalized, patient-tailored medicine, it could be useful to investigate in detail the mechanisms of neuroplasticity and neuroinflammation after thrombolysis and the relative differences with patients who did not receive this intervention.

There is a need to consider that the differences observed between TG and NTG could be related to inherent a priori differences between patients who received and did not receive the thrombolytic treatment. In this case, a matched analysis should be used. However, we did not perform it for the following reasons: As shown in Table 1, only two variables significantly differed between the two subgroups: age and Bamford classification. Despite being statistically significant, the age difference was slight (on average, only 3.6 years). In the Bamford classification, the TACI category resulted in a prognostic factor in both groups, whereas the LACI category was really rare in TG (only 0.7%), avoiding the possibility of performing a matched comparison on an adequate sample (Table 1). 

The findings of our study should be taken into account in light of its limitations, mainly referring to the use of testing a single specific algorithm for ANN. Despite already being used in previous studies [5,14,15,16], many other algorithms could be used [11]. The results of our ANN were then compared to those of a classical logistic regression, but not with those of other statistical approaches such as tree based models, which have previously been shown to outperform neural networks on tabular data also in the field of neurorehabilitation [5]. Another potential limit of our study is the sample size. Usually, machine learning needs a larger sample size. However, our sample size of 862 patients was similar to the average of the studies analyzed in a recent review about the use of machine learning to identify the prognostic factors of stroke neurorehabilitation, which was 891 subjects [11]. Furthermore, our study did not analyze other factors reported in some papers, such as Brunnstrom stage [25], parameters extracted from neuroimaging (structural and fMRI), EEG and genetics [11], or inflammatory biomarkers [26]. 

## 5. Conclusions

Despite the fact that thrombolysis is a powerful prognostic factor for stroke sequelae in the acute phase, it is not statistically significant for predicting neurorehabilitation outcomes in the subacute phase. However, our studies showed that the prognostic factors were partially different between subjects treated with or without thrombolysis. In particular, hypertension and epilepsy are two negative prognostic factors for subjects treated with thrombolysis, whereas dysphagia and the time from stroke to admission to a neurorehabilitation hospital were more important for patients not treated with thrombolysis. The outcome of these latter patients was also influenced to a greater extent by the severity of stroke at admission, as assessed by the Barthel Index. Many of the above factors were strictly intertwined, and it could have complicated the identification of powerful prognostic factors [5,17,18,19,21]. These, in addition to the high heterogeneity of stroke, may require supervised algorithms of AI [11], in which patients are previously classified according to their clinical histories until their admission to neurorehabilitation hospitals, and thrombolysis could be a first discriminant factor. Our results suggest the clinical need for paying particular attention to hypertension and epileptic events during the rehabilitation period for patients who underwent thrombolytic treatment, whereas for the other patients, there is a need to start neurorehabilitation as soon as possible. 

## Figures and Tables

**Figure 1 biomolecules-13-00334-f001:**
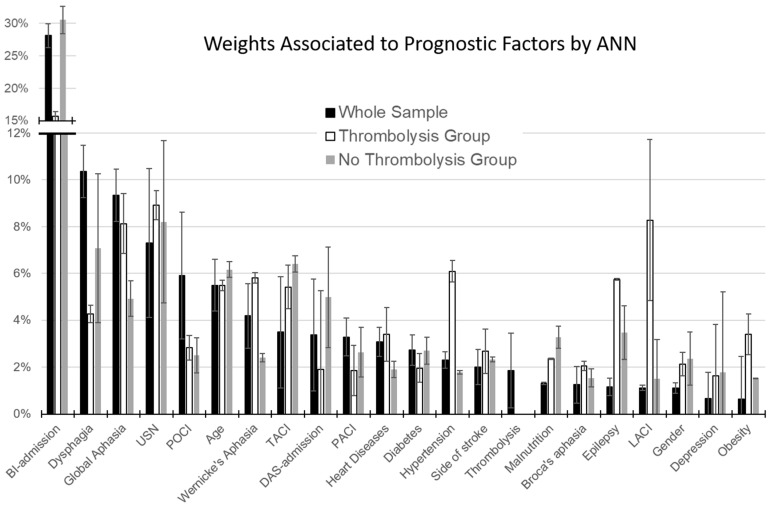
Percentage weights associated by Artificial Neural Network with the factors accounting for the predicted outcome (error bars represent the standard errors). Black bars refer to the whole sample of patients, white bars to those treated with thrombolysis (TG), and grey bars to patients not treated with thrombolysis (NTG).

**Table 1 biomolecules-13-00334-t001:** Demographical and clinical characteristics of the whole sample, the subgroup treated with thrombolysis (TG), and the subgroup not treated with it (NTG). Data are reported as mean ± standard deviation for age, time from stroke, and BI-scores and as percentage relative frequencies for all the other variables. The last column reports the *p*-values of the comparisons between subgroups performed using the U-test for ordinal and continuous variables and the chi-squared test for binary variables (Abbreviations: Time from stroke DAS: days at admission from stroke; USN: unilateral spatial neglect; BI: Barthel Index; PACI: partial anterior circulation infarction; TACI: total anterior circulation infarction; LACI: lacunar circulation infarction; POCI: posterior circulation infarction).

Demographical and Clinical Variables	Whole Sample (N = 862)	TG (N = 140)	NTG (N = 722)	*p*-Values
Age	70.3 ± 13.9	67.3 ± 14.4	70.9 ± 13.7	0.004
Gender	M: 52.4%	M: 54.3%	M: 52.1%	0.632
Time from Stroke (DAS)	18.5 ± 17.8	16.5 ± 14.9%	18.9 ± 18.3%	0.092
Side of stroke	R: 56.8%	R: 59.3%	R: 56.4%	0.524
Hypertension	70.8%	65.7%	71.7%	0.151
Heart diseases	45.6%	50.7%	44.6%	0.184
Diabetes	20.8%	15.7%	21.7%	0.107
Depression	38.1%	39.3%	37.8%	0.742
Epilepsy	5.5%	3.6%	5.8%	0.284
Dysphagia	72.3%	77.5%	71.2%	0.133
Malnutrition	3.8%	5.8%	3.5%	0.192
Obesity	14.4%	19.7%	13.3%	0.050
Bamford classification	PACI: 54.4%	PACI: 62.9%	PACI: 52.7%	0.027
	TACI: 21.6%	TACI: 30.0%	TACI: 19.9%	0.028
	LACI: 6.5%	LACI: 0.7%	LACI: 7.6%	0.002
	POCI: 17.0%	POCI: 5.7%	POCI: 19.1%	<0.001
Broca’s Aphasia	16.7%	20.0%	16.1%	0.254
Wernicke’s Aphasia	4.3%	6.4%	3.9%	0.173
Global Aphasia	13.6%	16.4%	13.0%	0.281
USN	22.3%	27.1%	21.3%	0.130
Emergency Transfer	11.3%	8.6%	11.8%	0.233
Deaths	2.2%	0.0%	2.6%	0.048
BI-admission	25.9 ± 27.2	22.6 ± 25.2	26.5 ± 27.6	0.187
BI-discharge	63.5 ± 33.0	60.8 ± 32.8	64.0 ± 33.0	0.360
Good Responders	40.7%	38.6%	41.1%	0.572

**Table 2 biomolecules-13-00334-t002:** Accuracy, Sensitivity, Specificity, and Area Under the ROC Curve of the ANN in identifying the correct outcome of patients.

Phase	Feature	Whole Sample	TG	NTG
Training	Accuracy	85.6%	85.5%	87.5%
	Sensitivity (good responders)	81.9%	68.0%	87.1%
	Specificity (other responders)	88.3%	85.5%	87.8%
Test	Accuracy	86.2%	85.1%	83.9%
	Sensitivity (good responders)	84.9%	80.8%	79.1%
	Specificity (other responders)	87.1%	87.8%	87.4%
Area Under the Curve		0.907	0.890	0.914

**Table 3 biomolecules-13-00334-t003:** Results of Binary Logistic Regression reporting the odd ratios (OR) and related *p*-values of variables that were entered into the model (reported in bold being statistically significant). For the variables that were not entered into the model, only the *p*-value is reported (* if the *p*-value of variables entered into the model was not <0.05, the *p*-value of the effect of removing the variable from the model was reported to clarify the statistical meaning). TG stands for thrombolysis group, NTG for no-thrombolysis group (BI: Barthel Index, USN: unilateral spatial neglect, PACI: partial anterior circulation infarction, TACI: total anterior circulation infarction, LACI: lacunar circulation infarction, POCI: posterior circulation infarction).

	Whole Sample	TG	NTG
Age > = 65	**OR = 0.33 (*p* < 0.001)**	**OR = 0.26 (*p* = 0.014)**	**OR = 0.32 (*p* < 0.001)**
BI-admission > 20	**OR = 14.17 (*p* < 0.001)**	**OR = 24.90 (*p* < 0.001)**	**OR = 15.1 (*p* < 0.001)**
Time from stroke < 15 days	**OR = 2.64 (*p* < 0.001)**	*p* = 0.594	**OR = 2.93 (*p* < 0.001)**
Right Hemisphere Stroke	**OR = 1.95 (*p* = 0.002)**	*p* = 0.403	**OR = 1.86 (*p* = 0.010)**
Dysphagia	**OR = 0.33 (*p* < 0.001)**	*p* = 0.416	**OR = 0.30 (*p* < 0.001)**
TACI	**OR = 0.01 (*p* = 0.011)**	**OR = 0.29 (*p* = 0.042 *)**	**OR = 0.40 (*p* = 0.036 *)**
Global Aphasia	**OR = 0.17 (*p* = 0.004)**	*p* = 0.239	**OR = 0.15 (*p* = 0.010)**
LACI	*p* = 0.867	**OR > 99 (*p* = 0.046 *)**	*p* = 0.905
Epilepsy	*p* = 0.756	**OR = 0.001 (*p* = 0.006 *)**	*p* = 0.588
USN	*p* = 0.086	*p* = 0.359	*p* = 0.203
Obesity	*p* = 0.394	*p* = 0.854	*p* = 0.215
Malnutrition	*p* = 0.337	*p* = 0.866	*p* = 0.375
Hypertension	*p* = 0.327	*p* = 0.154	*p* = 0.852
Heart Diseases	*p* = 0.875	*p* = 0.249	*p* = 0.968
Broca’s Aphasia	*p* = 0.586	*p* = 0.185	*p* = 0.694
Wernicke’s Aphasia	*p* = 0.828	*p* = 0.745	*p* = 0.842
Depression	*p* = 0.254	*p* = 0.474	*p* = 0.196
PACI	*p* = 0.759	*p* = 0.713	*p* = 0.657
POCI	*p* = 0.820	*p* = 0.814	*p* = 0.573
Accuracy	84.8%	85.3%	85.5%

## Data Availability

Data are available upon request from the corresponding author.

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
