# Peer review of "Application of an Artificial Neural Network to Identify the Factors Influencing Neurorehabilitation Outcomes of Patients with Ischemic Stroke Treated with Thrombolysis"

_biomolecules, 2023, doi:10.3390/biom13020334_

Round 1

Reviewer 1 Report

The authors attempt to use a neural network to decipher important prognostic features for neurorehabilitation in stroke patients treated with or without thrombolysis agents.  The  objective is interesting and the impact could be significant. However, there are several methodological clarifications that need to be made to ensure results match stated conclusions.

I realize the authors are re-using an existing data set. However, there needs to be a few more details summarizing patient inclusion processes,  specific inclusion criteria for the study, and basic protocols for when a patient was recommend for TG and when they were not.  Also, there needs to be more details on what TG protocol(s) were used at the study site.  Minimally, what was the most common standard of care protocol for TG?

The authors make the assumption that all explanatory variables are independent. Assuming all the explanatory variables are independent may not be true (for example, the cardiovascular variables are almost always clinically correlated).  I could not find where it was stated whether the authors check their assumption. The authors should check for excessive correlation or multicollinearity between explanatory variables to be sure. Corresponding results should be stated in the manuscript or in an appendix/supplement. 

The authors use a neural network. However, the sample size (especially for patients who received TG) is relatively small for a neural network.  Most neural networks need about 10,000 samples to be reliable. The lack of TG samples creates a large degree of class imbalance between TG and non-TG that the authors should explore and address.  Oversampling and undersampling or gradient boosting could be a possibilities. But beyond that, it is recommend that the authors perform multiple different classification algorithms: random forest, decision trees, Naive Bayes, support vector machine, logistical regression, etc. to compare their classification outcomes and corresponding feature weightings.  Those simpler methods may actually perform better than the neural network and provide some interpretability into the results.  Also, it would allow comparability between models to see which features were consistently picked by each method. 

The results illustrate that BI-admission weights extremely heavily on classification.  It is possible that this one feature may be used solely to do the classification.  If that's true, the remaining results are not valuable. The authors need to attempt recursive feature elimination OR minimally take one or two of the top factors and re-run the classification. This would provide important evidence to determine if one or two factors is overly biasing the classification results.

Pairing the approach with an unsupervised process  could also provide additional results clarity. Perhaps a t-SNE, PCA, or k-means clustering could find if there are clusters beyond the binary classes of interest.

For all machine learning and analysis, the authors need to provide which software, packages, and versions were used inside the Methods section.

The clinical context is very bare bones (too minimalist).  It is suggested the authors add more clinical context for their features and hypotheses in the Discussion for why said features are prognostic (or not).  What does this mean in terms of improving future clinical protocols?

Additionally, the written English is very difficult to read, even for an initial manuscript draft - the authors should have a native English speaker to assist in editing for clarity.  There are several places throughout where words out of place and/or missing. Moreover, there are several very long compound or complex sentences. Such sentences make the manuscript convoluted to read and grammatically incorrect.  The Introduction and Discussion is littered with such very long run-on sentences.

Author Response

We would like to thank the Editor and the Reviewers for their general positive judgments about our work and for their qualified comments that helped us to improve our work in the submitted revised version. In the following our point by point answers to explain how we have carefully taken into account the reviewers’ suggestions in the revised manuscript. We also reported the new or changed parts of the manuscript also in this document, between apices and in italic style.

REVIEWER 1: The authors attempt to use a neural network to decipher important prognostic features for neurorehabilitation in stroke patients treated with or without thrombolysis agents.  The  objective is interesting and the impact could be significant. However, there are several methodological clarifications that need to be made to ensure results match stated conclusions.

AUTHORS: Thank you for your general positive judgment about our work, and for the qualified comments. In the following, our point-by-point responses

I realize the authors are re-using an existing data set. However, there needs to be a few more details summarizing patient inclusion processes,  specific inclusion criteria for the study, and basic protocols for when a patient was recommend for TG and when they were not.  Also, there needs to be more details on what TG protocol(s) were used at the study site.  Minimally, what was the most common standard of care protocol for TG?

AUTHORS: The reviewers highlighted two interesting points. About the need of more details about the inclusion criteria, we have added the following paragraph:

“The inclusion criteria were: diagnosis of ischemic stroke confirmed by brain imaging (magnetic resonance imaging or computerized tomography), subacute phase of stroke, admission to neurorehabilitation hospital. The exclusion criteria were: previous cere-brovascular accidents, hemorrhagic stroke, subarachnoid hemorrhage, presence of other chronic disabling pathologies (i.e., severe Parkinson's disease, polyneuropathy, cancer, or limb amputation).”

About the need of more details about the common standard of care protocol for TG we have added the following paragraph:

“Guidelines and authorizations for of thrombolytic treatment administration are quite different among Countries: in Europe it is recommended its use within 4.5 hours from the acute event, with cautions against this use for severe stroke and people older than 80 years (in USA the limit of time was 3 hours, without the above recommended cautions) [1].”

REVIEWER 1: The authors make the assumption that all explanatory variables are independent. Assuming all the explanatory variables are independent may not be true (for example, the cardiovascular variables are almost always clinically correlated).  I could not find where it was stated whether the authors check their assumption. The authors should check for excessive correlation or multicollinearity between explanatory variables to be sure. Corresponding results should be stated in the manuscript or in an appendix/supplement. 

AUTHORS: Sorry, there was a misunderstanding. When we wrote “independent variables” we did not refer to the possibility that they were independent each other, but we referred to the Barthel Index score at discharge as “dependent variable” meaning it could depend on the other variables that we called independent, but we meant independent by the dependent variable. So, to avoid this misunderstanding, we have now removed the adjective “independent” when referring to the variable assessed at the baseline and being the input of our ANN.

REVIEWER 1: The authors use a neural network. However, the sample size (especially for patients who received TG) is relatively small for a neural network.  Most neural networks need about 10,000 samples to be reliable. The lack of TG samples creates a large degree of class imbalance between TG and non-TG that the authors should explore and address.  Oversampling and undersampling or gradient boosting could be a possibilities. But beyond that, it is recommend that the authors perform multiple different classification algorithms: random forest, decision trees, Naive Bayes, support vector machine, logistical regression, etc. to compare their classification outcomes and corresponding feature weightings.  Those simpler methods may actually perform better than the neural network and provide some interpretability into the results.  Also, it would allow comparability between models to see which features were consistently picked by each method. 

AUTHORS: We agree with the reviewer. However, the sample sizes used in studies applying ANN to neurorehabilitation are very variable. We have now added the following sentence to highlight this point:
“Another potential limit of our study is the sample size. Usually, machine learning needs a larger sample size. however, our sample size of 862 patients was similar to the average one of the studies analyzed in a recent review about the use of machine learning to identify the prognostic factors of stroke neurorehabilitation that was of 891 subjects [11].

Then, according to this comment of the reviewer, we have stressed this comparison in the paragraph titled

“3.3 Comparison between ANN and Regression Analysis

To confirm or not the findings of the ANN, a classical binary logistic regression was performed for the whole sample, for TG and for NTG. The results were partially similar to those obtained with ANN, with thrombolysis that remained out of the model for the whole sample (p=0.664). Then, BI-score at admission, age, and TACI classification resulted the three significant prognostic factors consistently in the three groups.

Similarly, to the findings obtained by ANN, DAS and dysphagia entered into the predictive model of NTG but not for TG (as well as stroke located in right vs. left hemisphere, not accounted by ANN). Epilepsy and LACI entered into the model of TG and not in that of NTG, in accordance with the findings of ANN. Hypertension and heart diseases remained out of the model but with lower p-values for TG than for NTG, reflecting the results of ANN. In addition, malnutrition remained out of the model, but with higher p-value for TG than NTG. The most important different result was observed for global aphasia, that entered into the model of NTG and not that of TG, on the contrary of what suggested by ANN.”

This comparison has been widely commented in Discussion, in particular with (but not only with) this paragraph:

“The regression analyses mainly confirmed the results of ANN. However, the former identified 7 significant variables for the whole sample and NTG, similarly to the 6 variables with a weight >5% identified by ANN for these groups. Conversely, ANN identified 9 variables with a weight >5% for TG, providing a more complex picture than that described by the only 5 significant factors of regression analysis. The regression analyses mainly confirmed the results of ANN. However, the former identified 7 significant variables for the whole sample and NTG, similarly to the 6 variables with a weight >5% identified by ANN for these groups. Conversely, ANN identified 9 variables with a weight >5% for TG, providing a more complex picture than that described by the only 5 significant factors of regression analysis.”

Finally, in Discussion we have also added the following sentence:

“Then, we compared the results of our ANN with a classical logistic regression, and not other statistical approaches such as tree based models, that already showed to be able to outperform Neural networks on tabular data also in neurorehabilitation field [5].”

REVIEWER 1: The results illustrate that BI-admission weights extremely heavily on classification.  It is possible that this one feature may be used solely to do the classification.  If that's true, the remaining results are not valuable. The authors need to attempt recursive feature elimination OR minimally take one or two of the top factors and re-run the classification. This would provide important evidence to determine if one or two factors is overly biasing the classification results.

AUTHORS: Thank you for this suggestion: this further analysis could allow us to verify the reliability of our results. In fact, we followed your suggestion and reported the following paragraph in Results section:

“Because BI-admission heavily weights on the results of ANN, a recursive confirmatory analysis was performed removing this factor from the input variables. Re-running the classification on the whole sample, the factors accounted for a percentage of importance >5% were: global aphasia (16%), USN (13%), time from stroke (12%), age (11%), dysphagia (9%), TACI (9%), Wernicke’s aphasia (5%). These results substantially confirmed those of the primary analysis.”

REVIEWER 1: Pairing the approach with an unsupervised process  could also provide additional results clarity. Perhaps a t-SNE, PCA, or k-means clustering could find if there are clusters beyond the binary classes of interest.

AUTHORS: All these adjunctive analyses could be very interesting, but we would say that they deserve a comparative specific study, whereas they could complicate this one (and also with a length longer than the manuscript allowed to this journal), especially for the readers of a journal as biomolecules. Then, our accuracy was similar to those of other similar studies also using other approaches. We have discussed it adding the following new paragraph:
“Our ANN accuracy of about 85% falls into the range of 74-95% reported by a recent review about the use in stroke rehabilitation of different machine learning approaches (including random forest, gradient boosting, support vector machine, decision tree and k-nearest neighbors) [11].”

We have however added to the limit of our study the following sentence to take into account the point risen by the referee:
“Then, we compared the results of our ANN with a classical logistic regression, but not with other statistical approaches such as tree based models, that already showed to be able to outperform Neural networks on tabular data also in neurorehabilitation field [5].”

REVIEWER 1: For all machine learning and analysis, the authors need to provide which software, packages, and versions were used inside the Methods section.

AUTHORS: We have now added this information writing in the methods section: “The ANN was developed in the Statistical Package for the Social Sciences software (SPSS) of the IBM, version 23.

REVIEWER 1: The clinical context is very bare bones (too minimalist).  It is suggested the authors add more clinical context for their features and hypotheses in the Discussion for why said features are prognostic (or not).  What does this mean in terms of improving future clinical protocols?

AUTHORS: The reviewer is right, we have now added the missed information adding the following new subsection in Material and Methods about the clinical context in which the study was conducted:

“2.2 Neurorehabilitation

Our neurorehabilitation ward is part of a hospital for subacute rehabilitation and it is formed by a wide gym, specific rooms for individual treatments, and bedrooms with two beds each one. The neurorehabilitation was planned for each patient by a pool of neurologists and physiatrists and administered by therapists 6 days a week, 3 sessions per day, each session lasting 1 hour. According to the needs of the patient, individual therapy could include physical therapy, cognitive therapy, neglect or speech therapy, specific therapy for swallowing, bowel, and bladder dysfunctions. All rehabilitation treatments began within 24 h from admission. Physiotherapy and language treatment continued throughout the hospital stay and the training for neglect lasted 8 consecutive weeks”.

Furthermore, we have added in Discussion the implication of our findings for improving future clinical protocols:

“Our results suggest the clinical need of deserving particular attention to hypertension and epileptic events during the rehabilitation period for patients who underwent to thrombolytic treatment, whereas for the other patients there is the need to start the neurorehabilitation as soon as possible.”

REVIEWER 1: Additionally, the written English is very difficult to read, even for an initial manuscript draft - the authors should have a native English speaker to assist in editing for clarity.  There are several places throughout where words out of place and/or missing. Moreover, there are several very long compound or complex sentences. Such sentences make the manuscript convoluted to read and grammatically incorrect.  The Introduction and Discussion is littered with such very long run-on sentences.

AUTHORS: A careful English revision has been performed throughout the whole manuscript.

Reviewer 2 Report

Tree based models often outperform Neural networks on tabular data. Why use ANN? What is the justification?

Author Response

We would like to thank the Editor and the Reviewers for their general positive judgments about our work and for their qualified comments that helped us to improve our work in the submitted revised version. In the following our answer to explain how we have carefully taken into account the reviewers’ suggestion in the revised manuscript. We also reported the new or changed parts of the manuscript also in this document, between apices and in italic style.

REVIEWER 2:

Tree based models often outperform Neural networks on tabular data. Why use ANN? What is the justification?

AUTHORS: Thank you for the general positive judgment about our manuscript. About this point, we agree with the reviewers. We have been invited by the journal Editorial staff of Biomolecules to submit a study on the Special issue titled “Advances in Drug Design and Development for Human Therapeutics Using Artificial Intelligence II”, so the use of ANN was a specific request of this special issue. However, to clarify this point also in our manuscript, and to take into account the correct point of view of the reviewer, we have added the last sentence in this paragraph of the Discussion of our manuscript:

“However, the findings of our study should be taken into account at the light of its limits, mainly referring to the use of testing one algorithm for ANN. Despite already used in previous studies [5, 14-16], many other algorithms could be used [11]. Then, we compared the results of our ANN with a classical logistic regression, but not with other statistical approaches such as tree based models, that already showed to be able to outperform Neural networks on tabular data also in neurorehabilitation field [5].”

Reviewer 3 Report

Thanks for the opportunity to review the manuscript. In it, the authors utilized an ANN to attempt to identify factors associated with the neurorehabilitation outcome when comparing patients who received IVT with those who did not. their results show that there are different explanative factors and confirm that IVT is not one of them. 

The text is well-written and reads quickly. The machine-learning (ML) approach to the issue of IVT as a prognostic factor is novel and well-applied. Thus the manuscript is worth publication.

Some comments aimed at aiding clarity are the following:

  1. In the methods section, some variables are adequately described for a readership not savvy in neurology. Still, others are left without a brief explanation (for example, epilepsy or the distinct types of aphasia). Additionally, the Bamford category is placed among binary variables even though it is not. 
  2. Please describe if the division of the data set into training and testing was randomized and the randomization method.
  3. Including a link to the code if possible and, if not, at least a brief description of the platform used for the ANN (coding language used, compilator, etc.). This comment also applies to logistic regression.
  4. The matter of global aphasia as a poor prognostic factor for rehabilitation is well-studied. It derives from the fact that the patient is incapable of receiving and following the instruction of the medical personnel.
  5. Please include a footnote in each table to spell out all abbreviations and clarify the numerical data format (for example, percentages of mean, median, sd, etc.)
  6. One possible explanation for the differences is the possibility of brain hemorrhage secondary to the use of IVT. Do the authors have any data regarding hemorrhagic complications in their sample?
  7. The main issue of this work is that the ANN classified the patients according to their outcomes after neurorehabilitation. Still, a priori patients who receive IVT are inherently different from those who don't. Therefore, the best approach would be to use matched controls for each IVT patient. The matching would have to be done based on all the significantly different variables in Table 1. If the authors could perform such matched case-control design, it would substantially increase the generalizability of the findings. Regardless, I understand the difficulties in doing so, being the main one that it would diminish the sample size available. This point is only marginally touched at the end of the conclusion section, and it deserves more attention. 

Author Response

We would like to thank the Editor and the Reviewers for their general positive judgments about our work and for their qualified comments that helped us to improve our work in the submitted revised version. In the following our point by point answers to explain how we have carefully taken into account the reviewers’ suggestions in the revised manuscript. We also reported the new or changed parts of the manuscript also in this document, between apices and in italic style.

REVIEWER 3: Thanks for the opportunity to review the manuscript. In it, the authors utilized an ANN to attempt to identify factors associated with the neurorehabilitation outcome when comparing patients who received IVT with those who did not. their results show that there are different explanative factors and confirm that IVT is not one of them. 

The text is well-written and reads quickly. The machine-learning (ML) approach to the issue of IVT as a prognostic factor is novel and well-applied. Thus the manuscript is worth publication.

AUTHORS: We would like to thank you for your very positive judgment about our work and for the further suggestions

REVIEWER 3: Some comments aimed at aiding clarity are the following:

  1. In the methods section, some variables are adequately described for a readership not savvy in neurology. Still, others are left without a brief explanation (for example, epilepsy or the distinct types of aphasia). Additionally, the Bamford category is placed among binary variables even though it is not. 

AUTHORS: We have better detailed some clinical aspect writing:

“…if there was a diagnosis of hypertension, heart diseases, diabetes, depression, epilepsy, dysphagia, malnutrition, obesity, Broca’s aphasia (related to deficits in speech and language production), Wernicke’s aphasia (related to deficits in language understanding), global aphasia (including both the previous two types of language deficits), unilateral spatial neglect (USN, related to deficits in reporting or responding to stimuli presented from the space contralateral to the lesion, often a right hemisphere lesion), and the category of Bamford classification. This latter variable refers to the anatomical type of stroke and were furtherly divided in 4 binary variables, in accordance with previous studies who dichotomized each one of these categories [3,5]: TACI (total anterior circulatory infarction), PACI (partial anterior), POCI (partial posterior), and LACI (lateral anterior).”

REVIEWER 3:

2) Please describe if the division of the data set into training and testing was randomized and the randomization method.

AUTHORS: According to this comment, we have added the following sentence:

The procedure uses random number generation during random assignment of partitions for subsampling cases between training and testing.”

REVIEWER 3:

3) Including a link to the code if possible and, if not, at least a brief description of the platform used for the ANN (coding language used, compilator, etc.). This comment also applies to logistic regression.

AUTHORS: We have added the required information with the following two new sentences:
“The ANN was developed in the Statistical Package for the Social Sciences software (SPSS) of the IBM, version 23.”

“All the statistical analyses, as well as the ANN, have been performed using SPSS (Statistical Package for the Social Sciences) software of the IBM, version 23.”

REVIEWER 3:

4) The matter of global aphasia as a poor prognostic factor for rehabilitation is well-studied. It derives from the fact that the patient is incapable of receiving and following the instruction of the medical personnel.

AUTHORS: We agree with the reviewer, and we have pointed out this explanation adding the following sentence in the Discussion:

“In general, the role of global aphasia as a prognostic factor of a poor outcome in rehabilitation is studied and linked to the reduced capacity of patient to understand and follow the instructions given during therapy sessions.”

REVIEWER 3:

5) Please include a footnote in each table to spell out all abbreviations and clarify the numerical data format (for example, percentages of mean, median, sd, etc.)

AUTHORS: Done in Tables 1 and 3, where the required information were missing. In particular the legend of Table 1 now is as follows:

“Table 1. Demographical and clinical characteristics of the whole sample, the subgroup treated with thrombolysis (TG), and the subgroup not treated with it (NTG). Data have been reported as mean ± standard deviation for age, time from stroke and BI-scores, and as percentage relative frequencies for all the other variables. The last column reports the p-values of the comparisons between subgroups performed using u-test for ordinal and continuous variables, and chi-squared test for binary variables (Time from stroke DAS: days at admission from stroke; USN: unilateral spatial neglect; BI: Barthel Index, PACI: partial anterior circulation infarction, TACI: total anterior circulation infarction, LACI: lacunar circulation infarction, POCI: posterior circulation infarction).”

The legend of Table 3 is as follows:
“Table 3. Results of Binary Logistic Regression reporting the odd ratios (OR) and the related p-values of the variables that entered into the model (in bold). For the variables not entered into the model, it was reported only the p-value (* if the p-value of variables entered into the model was not <0.05, it was reported the p-value of the effect of removing the variable from the model for clarifying the statistical meaning). TG stands from thrombolysis group, NTG for no-thrombolysis group (BI: Barthel Index, USN: Unilateral Spatial Neglect, PACI: partial anterior circulation infarction, TACI: total anterior circulation infarction, LACI: lacunar circulation infarction, POCI: posterior circulation infarction)”

REVIEWER 3:

6) One possible explanation for the differences is the possibility of brain hemorrhage secondary to the use of IVT. Do the authors have any data regarding hemorrhagic complications in their sample?

AUTHORS: This is a good explanation, but we did not observe any hemorrhagic complications during neurorehabilitation in the subgroup treated with thrombolysis. However, it should be considered that patients with hemorrhagic complications in acute phase were not included into our study that take into account only ischemic single stroke. About blood pressure and hemorrhagic event we have added the following paragraph in discussion:
“In general, brain circulation and brain pressure are two fundamental factors both for the risk of having a stroke as well as for the implications during subacute and chronic phases, including the risk of a secondary stroke. Hence, the correct management of the hypertension is mandatory for primary prevention of stroke recurrences and also during the stroke neurorehabilitation program. Our results showed as this aspect is particularly important in patients who received thrombolytic treatment. Clinicians may lower blood pressure but at the same time it is necessary to ensure an adequate perfusion of the brain during neurorehabilitation. The optimal balance between these two aspects should be modulated, also to favor neuroplasticity, on the basis of the type of stroke, comorbidities [20], and, as suggested by our results, also considering if the patient received thrombolytic treatment. It should be considered that our study excluded patients who had an hemorrhage after thrombolysis in acute phase, further studies should investigate this aspect and potential hemorrhagic complications in people treated with thrombolysis.”

REVIEWER 3:

7) The main issue of this work is that the ANN classified the patients according to their outcomes after neurorehabilitation. Still, a priori patients who receive IVT are inherently different from those who don't. Therefore, the best approach would be to use matched controls for each IVT patient. The matching would have to be done based on all the significantly different variables in Table 1. If the authors could perform such matched case-control design, it would substantially increase the generalizability of the findings. Regardless, I understand the difficulties in doing so, being the main one that it would diminish the sample size available. This point is only marginally touched at the end of the conclusion section, and it deserves more attention. 

AUTHORS: The reviewer is theoretically right about the differences a-priori between the two groups. However, the variables significantly different between the two groups, as reported in Table 1, are just age and Bamford categories. Despite statistically significant, the difference in the mean ages was lower than 3 years. About Bamford classification, TACI category resulted statistically significant in both groups (as highlighted in Table 3), whereas the LACI category was really rare in TG (only 0.7%) and not allowed a matched comparison. We have explained this point in Discussion writing:

“There is the need to consider that the differences observed between TG and NTG could be related to inherent a-priori differences between patients who received and those who did not receive the thrombolytic treatment. In this case, a matched analysis should be used. However, we did not perform it for the following reasons. As shown in Table 1, only two variables significantly differed between the two subgroups: age and Bamford classification. Despite statistically significant, the age difference was slight (on average of only 3.6 years). About Bamford classification, TACI category resulted a prognostic factor in both groups (Table 3), whereas the LACI category was really rare in TG (only 0.7%) avoiding the possibility to perform a matched comparison on an ad-equate sample.”

Reviewer 4 Report

Please explain the prognostic factor role of brain circulation and brain pressure in neurorehabilitation outcomes of patients at different  stage before and after stroke.

Author Response

We would like to thank the Editor and the Reviewers for their general positive judgments about our work and for their qualified comments that helped us to improve our work in the submitted revised version. 

REVIEWER 4: Please explain the prognostic factor role of brain circulation and brain pressure in neurorehabilitation outcomes of patients at different  stage before and after stroke.

AUTHORS: This point would deserve a wide discussion, but for the sake of brevity and in relationship to the length limit imposed by the journal we have tried to summarized this aspect adding the following paragraph and a new reference that could allow the reader to go in depth about this topic. The added paragraph is the following:

“In general, brain circulation and brain pressure are two fundamental factors both for the risk of having a stroke as well as for the implications during subacute and chronic phases, including the risk of a secondary stroke. Hence, the correct management of the hypertension is mandatory for primary prevention of stroke recurrences and also during the stroke neurorehabilitation program. Our results showed as this aspect is particularly important in patients who received thrombolytic treatment. Clinicians may lower blood pressure but at the same time it is necessary to ensure an adequate perfusion of the brain during neurorehabilitation. The optimal balance between these two aspects should be modulated, also to favor neuroplasticity, on the basis of the type of stroke, comorbidities [20], and, as suggested by our results, also considering if the patient received thrombolytic treatment. It should be considered that our study excluded patients who had a hemorrhage after thrombolysis in acute phase, further studies should investigate this aspect and potential hemorrhagic complications in people treated with thrombolysis”

Round 2

Reviewer 1 Report

The authors have significantly improved their manuscript by: adding necessary details on the methods (including patient selection, clinical data and clinic site descriptions, as well as technical model specifications); performing additional requested analysis to confirm results; and adding more clinical context and discussion to model interpretations.  Key technical concerns have mostly been adequately addressed.  Only one remaining technical request is made regarding the presentation of results in Figure 1 below.

Figure 1 is a pivotal figure with the main model results (feature importance of model prediction). Typically ML model feature importance is shown with corresponding predicted error (could be a confidence interval, standard deviation, standard error, etc.) with error bars, etc.  Having this is really important for interpretation. It can be easily added using built-in packages. On a more minor note, the clarity of Figure 1 could be improved by adding tick marks to separate the features on the x-axis.   The authors may consider a split y-axis given all but 2 features are less than 10% (which makes these less important features difficult to discern from one another). Another option is adding data labels above the bars or adding tabular data to the Appendix which illustrates the actual percentages.

The English clarity is much improved compared to the original submission. However, moderate English editing is recommended prior to final publication to deliver the highest quality product.

Author Response

REVIEWER: The authors have significantly improved their manuscript by: adding necessary details on the methods (including patient selection, clinical data and clinic site descriptions, as well as technical model specifications); performing additional requested analysis to confirm results; and adding more clinical context and discussion to model interpretations.  Key technical concerns have mostly been adequately addressed.  Only one remaining technical request is made regarding the presentation of results in Figure 1 below.

AUTHORS: Thank you very much for your positive judgment about how we have revised our manuscript.

REVIEWER: Figure 1 is a pivotal figure with the main model results (feature importance of model prediction). Typically ML model feature importance is shown with corresponding predicted error (could be a confidence interval, standard deviation, standard error, etc.) with error bars, etc.  Having this is really important for interpretation. It can be easily added using built-in packages. On a more minor note, the clarity of Figure 1 could be improved by adding tick marks to separate the features on the x-axis.   The authors may consider a split y-axis given all but 2 features are less than 10% (which makes these less important features difficult to discern from one another). Another option is adding data labels above the bars or adding tabular data to the Appendix which illustrates the actual percentages.

AUTHORS: Thank you for these suggestions to improve our figure. According to them, we have added the standard error to the bars to improve the interpretation of the result, we have also added, as requires, tick marks to separate the features on x-axis, finally we split the y-axis to increase the readability of smaller columns as required. 

REVIEWER: The English clarity is much improved compared to the original submission. However, moderate English editing is recommended prior to final publication to deliver the highest quality product.

AUTHORS: Thank you for the positive judgment about our English editing, we have further improved it in this second revision, the further editing process, if the study will be accepted, will correct if there are some remaining typos or awkward sentences.
